# A Framework for Constructing a Secure Domain of Sensor Nodes

**DOI:** 10.3390/s19122797

**Published:** 2019-06-21

**Authors:** Janusz Furtak, Zbigniew Zieliński, Jan Chudzikiewicz

**Affiliations:** Faculty of Cybernetics, Military University of Technology, 00-908 Warsaw, Poland; zbigniew.zielinski@wat.edu.pl (Z.Z.); jan.chudzikiewicz@wat.edu.pl (J.C.)

**Keywords:** wireless sensor networks, trusted platform module, security in IoT

## Abstract

Application of the Internet of Things (IoT) in some critical areas (e.g., military) is limited mainly due to the lack of robust, secure, and trusted measures needed to ensure the availability, confidentiality, and integrity of information throughout its lifecycle. Considering the mostly limited resources of IoT devices connected by wireless networks and their dynamic placement in unsupervised or even hostile environments, security is a complex and considerable issue. In this paper, a framework which encompasses an approach to integrate some security measures to build a so-called “secure domain of sensors nodes” is proposed. The framework is based on the use of the Trusted Platform Modules (TPMs) in wireless sensor nodes. It encompasses an architecture of sensor nodes, their roles in the domain, and the data structures as well as the developed procedures which could be applied to generate the credentials for the sensor nodes, and subsequently, to build a local trust structure of each node as well as to build a trust relationship between a domain’s nodes. The proposed solution ensures the authentication of sensor nodes and their resistance against unauthorized impact with the hardware/software configuration allowing protection against malware that can infect the software. The usefulness of the presented framework was confirmed experimentally.

## 1. Introduction

Application of the Internet of Things in some critical areas (e.g., military, production process monitoring and automation, humanitarian assistance and disaster recovery (HADR) scenarios) is limited mainly due to the lack of robust, secure, and trusted measures needed to ensure the availability, confidentiality, and integrity of information. Considering the mostly limited resources of the IoT devices connected by wireless networks and their dynamic placement in unsupervised or even hostile environments, security is a complex and considerable issue. 

To design appropriate security mechanisms for such IoT networks, it is necessary to consider that most of the IoT network nodes (devices) will be mobile and connected by wireless links with low bandwidth and a relatively small range to exchange data. These devices usually have low computing power and relatively small memory resources. The provision of adequate power sources for such devices is also very limited, and the use of power sources other than batteries or accumulators (e.g., on a battlefield) is rarely allowed.

Assuming that the IoT network would be a source of reliable data, the same security requirements for confidentiality, integrity, and availability should be met as for traditional systems. It is worth noting that these requirements should also be met by communication links used in the IoT. When designing security solutions for critical IoT systems, one also needs to consider the dynamic placement of the IoT devices in unsupervised or even hostile environments, and therefore, their exposure to unauthorized activities. For this reason, each node should have security mechanisms implemented to protect data stored in their resources (e.g., by encrypting these data) and mechanisms to ensure the integrity of these data (e.g., by using the hash function). In addition, each node should be equipped with mechanisms to monitor the state of the hardware and software configuration of the node to detect unauthorized modification of these elements, and procedures should be present to respond to such incidents, e.g., resetting node data, sending messages about such incidents to other nodes, immobilizing the node or even physically destructing it.

Starting from the above highlighted challenging context, we concluded that providing data and communication security in IoT systems for special applications requires guarantee of not only the availability, confidentiality, and integrity of information throughout its lifecycle, but also the authentication of users and devices by using appropriate cryptographic protocols. It is also necessary to ensure resistance to some attacks related to the IoT perception layer (especially physical attacks using a fake identity for malicious or collusion attacks, attacks on the confidentiality and integrity during data transit, routing attacks) and faults in IoT networks. To meet the described above postulates, we propose a framework which encompasses an approach to integrate some security measures to build a so-called “secure domain of sensor nodes”. The framework is based on the use of the Trusted Platform Modules (TPM) in wireless sensor nodes. It encompasses an architecture of sensor nodes, their roles in the domain, and the data structures as well as the developed procedures which could be applied to generate the credentials for the sensor nodes, and subsequently, to build a local trust structure of each node as well as to build a trust relationship between the domain’s nodes. The proposed solution ensures the authentication of sensor nodes and their resistance against unauthorized impact with its hardware/software configuration, allowing protection against malware that can infect the software. It also protects against some routing threats with a specific focus on the wireless sensor networks (WSNs) by limiting routing based on the Routing Protocol for Low-Power and Lossy Networks (RPL) only between authenticated nodes. Within the framework, we also developed the key management procedures for protecting confidentiality and with integrity the data which is stored at nodes or transmitted. When developing the framework, we focused not so much on the cryptographic protocols themselves, but mainly on the development of data structures and procedures to prepare the domain. The framework was constructed in such a way that it allows a relatively simple exchange of cryptographic methods.

The presented concept of a secure domain for sensor nodes was experimentally tested. A domain demonstrator covering several sensor nodes was prepared for the needs of the research, and some efficiency examinations of the procedures of the domain deployment phase were performed. 

The organization of the rest of the paper is as follows. Section 2 reviews recent literature related to security issues in the IoT for critical applications, suitable security solutions in WSNs, and cryptographic key distribution in the IoT systems. A review of our previous publications on the concept of a secure sensors domain is also provided in this section. Section 3 describes the idea of a security domain of sensor nodes, particularly, the architecture of the secure domain, the concept of sensor node protection, the composition of a sensor node, and the structures of data stored in the resources of the sensor node. In this section, we provide an overall description of several procedures which have been prepared to cover all stages of the life of the entire secure domain and all stages of the life cycle of each sensor node as well. In Section 4, we present procedures for creating a secure domain of sensor nodes which include the procedure of initiating a secure domain, i.e., preparing the first node to perform the role of Master in the domain, and the procedures for registering subsequent nodes in the domain restoring node data after power on, refreshing the node status, and node status confirmation are also presented. Section 5 describes the domain demonstrator and experimental results. Section 6 summarizes the analysis of wireless bandwidth consumption for the main procedures and compares the security evaluation of the proposed solution with common real-life threads and attacks on wireless sensor networks. Finally, Section 5 concludes the paper and discusses open problems.

## 2. Related Works

In Reference [1], a taxonomic analysis of IoT security issues was conducted from the perspective of the three key layers of the IoT system model: (1) perception (related to the physical IoT sensors/actuators to support data collection and processing); (2) transportation (related to the ubiquitous access environment for the perception layer); and (3) an application layer was given. As a result of this analysis, the most critical issues were highlighted. According to this analysis, the most vulnerable layer of the presented IoT system model is the perception layer, mainly due to the physical exposure of IoT devices deployed in open environments and to the very large hardware constraints that limit the implementation of effective security measures. The critical issues are, among others [1], (1) a lack of lightweight cryptographic algorithms and effective key management; (2) a lack of lightweight trust management mechanisms; (3) insecure routing protocols; and (4) a lack of lightweight anti-malware solutions.

Currently, wireless security relies mainly on cryptographic techniques and protocols that lie in the upper layers of the wireless network. At the application layer, the constrained application protocol (CoAP) has modified the widely employed datagram transport layer security to fit it into memory/energy constrained devices [2].

In Reference [3], the security and privacy issues of two important IoT technologies, namely, WSN and radio-frequency identification (RFID), are discussed, among others. In this work, it is pointed out that some of the potential threats in the RFID system are replay, synchronization, Radio Frequency (RF) eavesdropping, data alteration, etc. In the context of IoT, more security requirements are expected from the RFID system like mutual authentication, a key establishment for data confidentiality and anonymity. Currently, compared to RFID, WSN security has been raised to a much higher level. There are more and more solutions for WSN nodes (e.g., IRIS, SunSPOT) which are capable enough of performing symmetric and asymmetric cryptographic operations with hardware support. 

Real and suitable security solutions in WSNs such as the 802.15.4 standard, Zigbee, Bluetooth, TinySec MiniSec, Spins, SecureSens, AMSecure, and Sizzle were evaluated in Reference [4]. Some of these WSN networks offer secure modes which provide data encryption, frame integrity, and sequential freshness (such as the 802.15.4 standard and Zigbee) or authenticated encryption, such as TinySec and AMSecure. Although many of these security solutions can be used to go part of the way towards effectively securing a WSN, no one solution can provide all the necessary mechanisms for the special application of the IoT. In Reference [5], the research progress of IoT is reviewed by analyzing the security requirements and presenting the different privacy and security techniques in the IoT.

In Reference [6], secure data transmission for cluster-based WSNs (CWSNs), where the clusters are formed dynamically and periodically, is investigated. Two secure and efficient data transmission protocols are proposed by using the identity-based digital signature scheme and the identity-based online/offline digital signature scheme, respectively. 

Cryptographic key distribution is a critical stage in the implementation of network security in critical IoT systems. Key management is one of the most important mechanisms used to secure encryption keys and their safe distribution among sensor nodes. Recently, many key management schemes have been proposed, each with its particular strengths, weaknesses, and applications under certain circumstances. The problem of key distribution in the IoT networks was recently investigated broadly by many researchers, e.g., [7,8,9,10,11,12,13]. For instance, in Reference [7], a key management protocol with a hybrid key management technique was proposed to meet the increasing resource-consuming security demands by asymmetric key primitives. In the proposed key management technique, symmetric and asymmetric key primitives are used at different levels of the hierarchy of sensor nodes. When applying the identity-based key management technique at the cluster head level, this solution consumes significantly more computational resources. The resource requirements of the pairing algorithm when implemented on the advanced RISC machine (ARM) processor using pairing functions were studied in Reference [8]. In Reference [9], the TinyPBC algorithm, which provides identity-based key management without interaction between participating nodes, was studied.

In Reference [10], a novel key management protocol (KMP), which integrates implicit certificates with the standard elliptic curve Diffie-Hellman exchange and performs authentication and key derivation, was presented. Although the proposed KMP guarantees maximal airtime savings concerning conventional approaches, robust key negotiation, and efficient protection against replay attacks, the airtime consumption requirement for the exchange of multiple messages and certificates remains significant. This causes major consequences due to the significant latency in the authentication protocol when running over a typical low-rate communication channel and the considerable power consumption.

In Reference [11], a cryptographic key management protocol based on the Identity-Based Symmetric Keying (IBSK) scheme was proposed. This protocol uses only two symmetric keys pre-deployed at each sensor and supports the ejection of the compromised nodes. Although the energy consumption overhead introduced by this key management scheme is remarkably low, this protocol is only suitable for network architectures in which only sensor-to-gateway secure sessions are allowed.

In Reference [13], a procedure using asymmetric and symmetric cryptography that reduces the memory requirement was proposed. The efficient Quantis Random Number Generator is used to provide symmetric keys with full entropy. The method of delivering symmetric keys (so-called session keys) to nodes wishing to implement a safety data exchange is presented.

A novel approach that was developed to provide secure group communication specifically to support dismounted warfighters at the tactical edge was presented in Reference [14]. The protocol supports a variety of requirements such as dynamic authorization and de-authorization of clients. This paper proposed a novel and secure key rotation mechanism based on multicast key dissemination for tackling the distribution of group encryption key updates at the tactical edge.

Another issue is related to the security of the IoT for critical applications, since ensuring system security is a necessity to gain trust. Usually, the Certification Authority (CA) is used as the third party of trust. Unfortunately, for IoT network nodes, due to the limited resources of these nodes and the scalability problems, it is challenging to develop effective procedures for CA utilization. For this reason, IoT nodes should rather be organized as separate groups (clusters) of cooperating nodes. A separate security system (the secure domain of sensors) should be built in each cluster. In such a cluster, trust structures should be created locally. To build a local trust structure in each safe domain of sensors, a Trusted Platform Module (TPM) can be used. This should be a necessary component of every node included in the safe domain of sensors [15,16,17]. This approach also requires the development of procedures for building trust between cooperating clusters.

The first implementation of the secure domain of sensor nodes, using our idea, was presented in Reference [15]. In this solution, each sensor node uses asymmetric cryptography supported by the TPM module to protect transmission and to protect the stored data in the resources of the sensor nodes and the authentication procedures. The result of using asymmetric cryptography was a large limitation of the domain size. The storage resources of sensor nodes allowed a domain containing no more than 11 sensor nodes to be built. The foundation of our next solution, presented in Reference [16], was symmetrical cryptography, which was also supported by TPM v1.2 [18]. In this solution, subsequent sensor nodes were registered in the domain without preliminary preparation. In the registration procedure for sensor nodes, a wireless link was used, and the procedure for registering subsequent nodes began with sending (in plain text) the public key of the M node through this link. This key was easy to intercept and then it was relatively easy to register a false sensor node in the domain. This approach made node M a very vulnerable point of the domain, and the nodes could be easily attacked before and during their registrations in the domain.

The initial concept of a secure domain solution including the phase of pre-preparation of nodes for work in the secure domain is presented in Reference [17]. In Reference [19], the procedures of the pre-preparation phase of sensor nodes are described. The paper describes the data structures and procedures performed in the phase of pre-preparation of sensor nodes in detail.

The contribution of the current study covers the following issues:A new data protection and distribution of cryptographic material between domain nodes in the phase of deployment and regular work of the sensors’ domain;The domain deployment procedures, in particular, the procedures for initiating the domain, the registration node in the security domain, restoring node data after powering on, refreshing the node status, and node status confirmation;An experimental test of the domain’s demonstrator for:○A wireless bandwidth consumption;○Evaluation of the domain resistance for the most important attacks;○Performance evaluation.

## 3. The Concept of a Security Domain of Sensor Nodes

### 3.1. Secure Domain Architecture

To build a secure network of IoT nodes, one should take into account the security observations described in the previous section and implement the following postulates:Wireless sensor nodes (**Sensor node**: Element of the sensor network, which includes at least the measuring element (sensor), a microcontroller, and a communication module that allows the transfer of measured data through wireless connections.**Sensor**: Measuring component of the sensor node) constitute the cluster of sensors (domain) in which security mechanisms are implemented;The resources of each node are protected by cryptography;Data transmission between cluster nodes and between clusters is protected by cryptography;Each cluster uses the local trust structure and a secure procedure for generating and distributing cryptographic keys;Each sensor node must be registered in the domain before it can start working in the domain. Registered nodes in the domain before they begin their activities must be authenticated;The necessary hardware component of each sensor node is a TPM.

From a security perspective, each secure domain of sensor nodes is autonomous, i.e., all procedures related to the creation of trust structures with the generation, distribution, and renewal of keys are performed internally in a secure domain. All necessary data for these procedures are stored in the protected resources of the domain sensor nodes. The cluster structure in which a secure domain of sensor nodes can be created is similar to that shown in Figure 1.

Cluster objects (sensor nodes) exchange data using wireless links. An example of such an object can be a soldier or an unmanned aerial vehicle (UAV) equipped with measuring sensors. The object’s task is to collect data from its measurement sensors, pre-process them, and securely transfer aggregated data to another sensor node.

The cluster nodes form two interpenetrating domains [15] (each cluster node belongs to both domains):**Security domain**. The domain of security is created by all sensor nodes registered in the domain. At a given moment, there is exactly one sensor node in the security domain that acts as a Master (M node). Such a node is the authority of security in the domain. Its resources store the description of the domain and the descriptions of all sensor nodes forming the domain and the necessary data to authenticate other nodes in the domain.The other nodes in their resources store a copy of the secure domain description obtained from the M node and act as a Replica (R node). In case of damage to the node acting as a Master, the procedure of selecting a new M node from correctly functioning R nodes is initiated.To secure transmission in the security domain between a given sensor node and the M node, a symmetric key (NSK—Node Secure Key) is used, which is known only for these two nodes. For this reason, the network creates star topologies in the security domain, regardless of the transmission medium used (Figure 2A)In the domain, diagnostic procedures are also performed to detect nodes that are working incorrectly. To secure transmission in these procedures, a symmetric key NDK (Node Diagnostic Key) common to all domain nodes is utilized.Each domain sensor node (both M nodes and R nodes) is equipped with measuring sensors and is a natural source of sensor data.**The domain of protected transmission**. In this domain, one of the sensor nodes is the recipient (sink node) of data from the other nodes of the domain and acts as the Gateway (G node) in this domain. Both the M node and R node can perform a Gateway role. The role of the G node is also to securely transfer sensor data coming from the domain to recipients working in other domains. (Figure 2B).

### 3.2. The Concept of Protection for Sensor Nodes

The creation of a secure domain of sensor nodes results from the fact that such a domain will perform its tasks in an unfriendly environment exposed to unauthorized impact on individual domain nodes. This means that every element of the domain should be pre-prepared before starting normal operations in this hostile environment. It is assumed that the preparation procedures of each sensor node should be performed in a safe and controlled environment outside the area of regular sensor node operation. This approach allows the secure preparation of further sensor nodes for work in the domain, regardless of the regular domain operation.

The task of pre-preparing the sensor nodes of the domain is carried out by the base node (B node). This node is also responsible for generating the Domain Key (DK) for the domain, defining the domain description parameters, generating identifiers for the future sensor nodes of the domain, and defining the transmission parameters for the wireless link used by the domain nodes. From the point of view of security procedures, the B node is not a member of the secure domain of sensor nodes. It works only in a safe area.

It was assumed that each node of the sensor domain would use its local trust structure. This trust structure will be used to secure data stored in the resources of a given node, as well as to build a trust relationship with other nodes belonging to the same security domain. For this purpose, a TPM is used, which is a necessary component of every sensor node.

The local trust structure is based on the hierarchy of asymmetric keys (e.g., Algorithm Rivest-Shamir-Adleman (RSA) 2048). At the top of the hierarchy is the Endorsement Key (EK), which, for a given module, can be generated only once. It is not possible to modify or reset this key. It is also not possible to read the private content of this key. The next keys in the hierarchy are the Storage Root Key (SRK) and the Domain Key (DK). The SRK key is generated during the take over ownership procedure of the TPM. During this procedure, a secret (a string of characters) is determined, which later will be necessary to perform authorized actions of the sensor node in the security domain. The DK key is common to all nodes of the secure domain and is used for only the first exchange of symmetric keys between domain nodes. The DK is distributed between the domain nodes using a specially prepared secure procedure. The DK distribution procedure between domain nodes is described in detail in [19,20]. The private part of this key is never available in plain text form outside of the TPM module, and the public part of this key, although available in plain text outside the TPM module, is never transmitted between the nodes in an open form. The DK can be treated as the identifier of the secure domain of the sensor nodes, and the EK can be treated as the identifier of the sensor node in the domain.

Access to the NVRAM of the TPM module is protected by TPM mechanisms that use the SRK of the module. For this reason, the sensor node identifiers, symmetric keys used by the node, and addresses of nodes acting as the Master and Gateway nodes in the domain, as well as the transmission parameters of the wireless link used by the nodes in the domain are stored in this memory. The other sensitive data that are stored in the EEPROM of the sensor node are cryptographically protected using a symmetric NK. This key is one of the keys stored in the NVRAM of the TPM.

#### 3.2.1. The Architecture of the Sensor Node

The sensor node is a mobile device. The XBee interface is used to exchange data inside the sensor node domain. The LoRa interface allowing data transmission over longer distances is intended for use by the sensor node to exchange data with other domains in a situation where a given sensor node acts as a Gateway. The component of the sensor node that processes data is the Arduino microcontroller. The TPM is an obligatory component of the sensor node. Also, each sensor node can use one or more measuring sensors. A block diagram of the sensor node is shown in Figure 3.

#### 3.2.2. Data Stored in the Resources of the Sensor Node

Each sensor node should be able to perform all tasks provided for nodes in the domain, and in particular, it should perform the role of Master, Gateway or both. Therefore, the resources and capabilities of each sensor node should be similar. The NVRAM of the TPM and the EEPROM of the Arduino microcontroller are intended for storing sensitive sensor node data. The NVRAM of the TPM is designed to store the sensor node’s own data, including the local trust structure of the sensor node, the necessary data for domain node authentication, and cryptographic keys which are utilized to protect the data stored in the EEPROM of the node and to secure the data transmitted from that node. The microcontroller EEPROM is intended to store the description of the sensor node domain and the descriptions of registered nodes in the domain. The RAM stores the status data of registered nodes in the domain. An example of data stored on the M node and R node is shown in Figure 4.


**The individual data of the sensor node include**
Asymmetric keys (EK, SRK, and DK), which creates local trust structure;NTAG—a unique tag generated for the node during the node preparation procedure, which is intended for one-time authentication of the node in the registration procedure of this node in the domain;N_ID—a sensor node identifier in the domain;Symmetric keys: NK—for encrypting the data stored in the EEPROM of the node (the key is used only internally by this node), NSK—for encrypting data in authenticating procedures of the node in the domain (the key is known only to this node and the M node), NTK—for encrypting transferred measurement data from the sensor node to the G node (the key is common to all nodes of the domain),NDK—for encrypting data in diagnostic procedures in the domain (the key is common to all nodes of the domain);Addresses in the used wireless network: NAD—own address, MAD—address of the M node, GAD—address of the G node;Parameters of the wireless network used in the domain:PAN_ID (ID of the Personal Area Network)—the identity of the wireless network (XBee) used in the sensors’ domain,CH—operating channel in a wireless network (XBee) used in the sensors’ domain,PIK (PAN Interface Key)—symmetric key used by the XBee module, common to all nodes in the PAN;Key_Desc—description of keys stored in the TPM non-volatile memory.



**Security domain description:**
DN—sensors’ Domain Name;RN (Role of Node)—defines the role of the node in the resources of which the description is stored {M, R, M+G, R+G};VER—the version number of the domain description;Periods (i.e., Period of Replication (PR), Period of Non-success Replication (PNR), and Time of data validity (TDV)) associated with the operation of R nodes.


**Description of domain nodes.** The description of each node consists of two parts.

The first part (in EEPROM) includes the following data:N_ID (Node ID)—ID of the sensor;RN (Role of Node)—the role filled by the node in the domain {M, R, M+G, R+G};NSK (Node Security Key)—symmetric key of a registered node in the security domain; it is used to encrypt the data sent between this node and M node;NAD (Node ADdress in the wireless network)—64-bit address of XBee module used by the node.

The second part (in RAM) consists of the following status data:N_ID (Node ID)—ID of the sensor;Stat—the status of the node; it can take one of the values:when a node description is stored on the MASTER node  {non-active, active, active non-confirmed},when the node description is stored on REPLICA   {ActiveNonConfirmedReplica};Time—a moment of the last and the effective transmission.

#### 3.2.3. Protection Used in Sensor Nodes

The mechanisms and security solutions for all types of sensor nodes (i.e., M node and R node) used in the sensor’s secure domain are very similar. These include the following:The TPM module is an obligatory element of the hardware configuration of each node;The TPM module supports the procedures of creating and using a local trust structure, generating symmetric keys, and determining SHA-1 and HMAC hashes;Cryptographic keys used by the sensor node, and other sensitive node data (among others N_ID, PAN_ID, CH, and PIK) are stored in the non-volatile memory of the TPM;Sensitive data regarding the secure domain stored in the EEPROM of the sensor node are protected by cryptography with the NK key stored in the TPM;The Platform Configuration Registers (PCRs) of the TPM enable the protection of the software and hardware of the sensor node against unauthorized modification.

### 3.3. Procedures in a Secure Domain

The necessary condition for building a secure system is to design security solutions for the software and hardware components of such a system. A crucial element is also the development of safe procedures for creating and using such a system and withdrawing it from an operation. For the needs of the secure domain of sensor nodes, several procedures have been prepared that cover all stages of the lifecycle of each sensor node as well as all stages of the life of the entire secure domain. These include the pre-preparation phase, the deployment phase, and the phase of regular domain work.

The node pre-preparation phase aims to create a local trust structure in the resources of such a node, generate symmetric keys used only by this node and transfer from the B node transmission parameters of the wireless link used by the domain nodes. For this purpose, the node being prepared must be directly connected to node B using the serial link. Symmetric keys (i.e., NK to protect data stored in node resources, NSK to secure data exchange in the domain, and the One Time Key (OTK) key for cryptographic protection of data transfer only during the node preparation procedure) and the first two elements of the trust structure (i.e., EK and SRK) are created locally by the node. The remaining data must be obtained from the B node. In order to obtain the DK to the B node, the public part of the SRK key of the node being prepared is sent first. This part of the SRK is used by the TPM of the B node to create a migration blob containing the protected DK. In response, the migration blob is sent to the node being prepared, and on this basis, the DK is added to the local trust structure of this sensor node. In the next step, the public part of the DK is used to pass the symmetric OTK to the B node. This key is used to protect the transfer of wireless link parameters from the B node. If the node being prepared is to serve as a Master, additionally from node B, the initial contents of the domain description and node description are sent. This exchange is also protected by the use of the OTK.

The purpose of the deployment phase is to initiate a node that will act as a Master and register of subsequent R nodes in the domain. The first nodes registered will play the Gateway role in the domain. The completion of M node initialization is tantamount to the constitution of a secure domain. During the initiation of the M node in its resources, an NTK is generated to protect sensor data transmission to the G node and NDK to protect data transmission during diagnostic procedures. The NTK and NDK are common to all registered nodes in the domain. To protect the data transmission between the M node and the given R node, the NSK of the R node is used. This key is known only for the M node and a given R node. In the first step of registering the R node in the domain, its NSK bound with the public part of the DK is sent to the M node. The M node in its resources stores this key in the description of the registered node. In the next steps, the M node sends the NTK and NDK, the G node address, and a copy of the domain description and descriptions of the domain nodes to the node being registered. During these activities, a wireless link is used, and the NSK of the R node is used to protect data exchange. After registration, the node is ready for regular operation in the domain.

In the regular phase of domain operation, a symmetric NTK key is used to protect sensor data sent to the G node. In the course of performing diagnostic procedures, which aim to assess the integrity of a secure domain and the suitability and reliability of domain nodes, the symmetric NDK key is used to protect transmission. Diagnostics procedures are initiated by the M node at fixed time intervals or by other domain nodes after detecting a lack of communication with a node acting as a Master or Gateway. If diagnostic procedures detect the absence of the M node or G node or the loss of credibility by these nodes, node selection procedures that will take over the Master or Gateway roles are initiated. Data exchange during diagnostic procedures is protected using a symmetric NDK key.

The M node counts independently for the NSK and NDK and the size of the data transferred using these keys. If the volume of transferred data exceeds a fixed value, the renewal procedure for the given key is initiated. The NSK key being renewed is generated by the node that owns the key and is then sent to the M node—data transfer is protected by the “old” NSK key. The new NDK key is generated by the M node and is passed to all registered nodes in the domain. The “old” NDK key is used to protect the exchange of data during the renewal of the NDK key. The size of the sent sensor data is counted by the G node—these data are protected using the NTK key. After exceeding the determined volume value of the transferred data using the NTK key, the G node initiates the renewal procedure for this key—an old NTK key is used to protect the transmission.

The lifecycle of the sensors’ secure domain consists of the following phases:**The pre-preparation phase**—activities related to the preparation of the sensor nodes for work in the sensors’ domain. The procedures of this phase are performed in a safe and controlled space outside the area of the regular operation of sensor nodes. In this phase, the most important tasks are performed by the base node (B node), which is a source of credentials for sensor nodes prepared to work in a secure domain. This phase includes the following procedures:1.The procedure for initiating the Base node (B node);2.The procedure for preparing a sensor node for registration in the security domain.**The deployment phase of solutions**—activities related to the creation of the secure domain of sensor nodes and the registration of new sensor nodes in the domain. The procedures of this phase can be performed in the area of the regular operation of sensor nodes:3.The procedure for initiating M node—the creation of data structures in the node resources, which act as the Master in the domain;4.The procedure for registering a sensor node in the security domain;5.The procedure for restoring the M node and the entire description of the security domain—recovery and verification of data after power on the M node;6.Procedure for refreshing the node status—the procedure is started by the M node to check the current state of the R node;7.The procedure for node status confirmation—the process starts when the R node is powered up.**The phase of regular domain work** while maintaining safety rules (security policy)—activities related to secure exchange of the data between the sensor nodes in the domain and among the domains of sensor nodes:8.The procedure for sending data from the R node to the G node;9.The procedure for sending data from the G node outside the domain;10.The diagnostic procedure for the domain;11.The procedure for electing a new M node;12.The procedure for removing nodes from the domain;13.The procedure for electing a new G node;14.Procedure for renewing the NSK;15.Procedure for renewing the NTK;16.Procedure for renewing the NDK.

In this paper, attention is focused on activities included in the second phase, i.e., on the security mechanisms for deploying the sensors’ domain.

## 4. Procedures for Creating a Secure Domain of Sensor Nodes

The task of the preparation phase is to generate the credentials for the prepared sensor node and to put these data in the resources of this node. A set of credentials is necessary for the correct execution of the registration procedure of a given sensor node in a secure domain. Such a dataset includes a trust structure for the node, cryptographic keys, and other data required by the registration procedure over which the wireless link is used. After the preparation phase, the contents of the node’s resources are similar to the R node shown in Figure 5.

It was assumed that the first prepared sensor node for the domain would be a node whose task would be to initiate a secure domain and perform the role of Master in this domain immediately after completing the initialization of a secure domain. For this reason, during the procedure of preparing such a sensor node in its EEPROM memory, a domain description and identifiers N_ID for nodes that will be registered in the domain in the future were additionally written. The content of such a node is shown in Figure 5. (data generated additionally only for the M node is highlighted in yellow). 

The steps of the domain creation phase include the procedure of initiating a secure domain, including preparing the first node to perform the role of Master in the domain and the procedure for registering subsequent nodes in the domain. After disconnecting the power supply, the status data of the sensor node is lost, and when the node is switched on again, it is necessary to restore the status data. To perform these tasks, procedures for the common node and the node acting as the Master were developed.

### 4.1. The Procedure for Initiating the M Node

This procedure is intended to finish configuring the node that will play the MASTER role in the security domain. Completion of this procedure is equivalent to the formation of a secure domain of sensor nodes. The node should be prepared early, which means that the procedure for preparing the node for registration in the security domain has already been completed. During this procedure, the contents of the NTK and NDK fields are established in the NVRAM of the TPM, the description of this node is written in the first position of descriptions of the domain nodes, and the content of fields in the node’s RAM is refreshed.

The sequence diagram for the M node initialization procedure and the data stored on the M node after the procedure are shown in Figure 6 (the effect of initialization procedure for M node is highlighted in yellow).

### 4.2. The Procedure for the Registration Node in the Security Domain

This procedure is intended to register the sensor node in the security domain. The data from the NTAG, N_ID, NSK key, NAD, and SQ of the R node, which were generated during the node preparation procedure, are bound using the public part of the DK and are then sent to the M node. The M node creates the description of the registered node and saves it in its EEPROM. It is assumed that the R node registered in the security domain as the first node will play the GATEWAY role in the domain of transmitting sensory data. It results in the generation of an NTK key for the domain. As a confirmation of registration, the R node in the security domain M node sends the following data to the R node: the updated role (RN) in the security domain, the gateway address (GAD), and the key (NTK+IV) used in the domain of transmitting sensory data. During the registration procedure, the wireless XBee link is used. The final stage of registering the R node involves performing the backup of the security domain description in the resources of the R node. The sequence diagram of the procedure for registration of the sensor node in the domain is shown in Figure 7.

The result of the node registration procedure in the security domain is the update of data in the M node resources and the resources of the registered node.

On the M node side, this update consists of entering the designation of the role performed by the node (i.e., R or R + G) to the appropriate entry in the domain description, the value of its NSK key, and the address (NAD) of that node. If the node being registered is a node that will act as a Gateway, its NAD address is also saved in the GAD field and will be passed as the gateway address to all other registered nodes. Additionally, for each registered node, its status data are updated in the RAM of the M node.

On the side of the registered node in the NVRAM of the TPM, the values of NTK and NDK and the GAD address are saved. Also, a copy of the domain description is saved in the EEPROM of the registered node. All of this data are obtained from the M node.

The contents of the M node and the first registered node in the domain are shown in Figure 8 and after the registration of the second node in Figure 9. In both drawings, the updated data are highlighted in yellow.

Detailed descriptions of the most important steps of the procedure (in Figure 7, the numbers in parentheses (e.g., (3)) preceding the descriptions indicate the numbers of the individual steps) are as follows:

**(3) Generate node registration request**—a node registration request sent to the M node is also a request to send an NTK key back to the R node. Do the following:Prepare the ***transfer_key_req*** packet (Figure 10) containing the following data: N_ID, NTAG, NSK key, NAD, and SQ of the registered R node. The packet is bound using the public part of the DK (the blob includes the the bound data and its length (4 bytes)),Send a ***transfer_key***
***req*** packet from the R node to the M node.

**(5) Update the R node description in the M node EEPROM.** Do the following:On the M node, unbind the data from the ***transfer_key_req*** packet using the private part of the DK and verify the correctness of the NTAG field. If it is not correct, print the message and stop the procedure,On the M node, prepare the R node description, and then encrypt this description using the NK and the IV vector of the M node and update the R node description in M node resources. The fields of the description should have the following values:N_ID = the content remains unchanged (the field is not encrypted);RN = “R + G” if it is the first R node registered in the security domain;RN = “R” if it is not the first R node registered in the security domain;NSK and IV = the NSK key and IV vector gathered from the ***transfer_key_req*** packet;NAD = NAD gathered from ***transfer_key_req*** packet;Stat = Active (0);Time = current time.Increase by 1 version number (VER) and update the domain description.Send a confirmation of registration to node R (***transfer_key_ans*** packet (Figure 11**)**). The confirmation contains N_ID, RN, NTK, IV, GAD, SQ and is encrypted using the NSK and IV vector of the R node. If the node is the first registered node in the security domain, the NAD address gathered from the ***node_description_req*** packet is stored in NVRAM of TPM as the gateway address (GAD) in the domain of transmitting sensory data. In another case, the GAD is acquired from NVRAM of the TPM.

**(7) Acquire the NTK key**. Do the following:On the R node, decrypt the data from ***transfer_key_ans*** packet using the NSK and verify the correctness of the SQ field. If it is not correct, print the message and stop the procedure;Based on the contents of the ***transfer_key_ans*** packet, update the RN, NTK, IV, GAD, and SQ fields, respectively, in NVRAM of TPM and RAM of the R node.

**(10) Acquire the NDK key**. Do the following:Send a request of diagnostic key (***diagnostic_key_req*** packet (Figure 12**)**) from the R node to the M node. The sequence number (SQ) is increased by 1. The packet is encrypted using the NSK key of the R node.On the M node, prepare a diagnostic key and send it to the R node (***diagnostic_key_ans*** packet). The SQ from the ***diagnostic_key_req*** packet is increased by 1. The packet is encrypted using the NSK key of the R node;Based on the contents of the ***diagnostic_key_ans*** packet, update the NDK, IV, and SQ fields, respectively, in NVRAM of TPM and RAM of the R node.

**(13) Acquire a copy of the domain description**. Do the following:Send a request of the diagnostic key (***domain_descr_req*** packet (Figure 13)) from the R node to the M node. The SQ is increased by 1. The packet is encrypted using the NSK of the R node;On the M node, prepare a domain description to backup (***domain_descr_ans*** packet) and send it to the R node. The SQ from the ***domain_descr_req*** packet is increased by 1. The packet is encrypted using the NSK of the R node;Based on the contents of the ***domain_descr_ans*** packet, update the DN, VER, PR, PNR, TDV, and SQ fields, respectively, in the EEPROM and RAM of the R node. On R node, the field RN in the domain description is updated to the RN of the node.

**(17) Acquire a copy of the description of the domain nodes**. The number *N* of transferred descriptions is limited by the size area of the R node EEPROM or by the number of descriptions on the M node. Only one full description of the node is transferred in one round of using the ***back_up_node_req*** packet and ***back_up_node_ans*** packet (Figure 14). If the descriptions of the nodes are not filled (this applies to unregistered nodes so far), in one ***back_up_node_ans*** packet, it is possible to send no more than nine incomplete descriptions of nodes. Do the following repeatedly:Send a request of the *i*th (i belongs to the range <1; N>) node description (***back_up_node_req*** packet) from the R node to the M node. The SQ is increased by 1. The packet is encrypted using the NSK of the R node;On the M node, prepare the node description (***back_up_node_ans*** packet) and send it to the R node. The received SQ is increased by 1. The Num is a serial number of the first requested node from the ***back_up_node_req*** packet. The packet is encrypted using the NSK of the R node. If the requested data concerns a registered node, then a “0” type packet containing a full description of this node is sent as an answer. If the requested data does not apply to the registered node, then the type “1” packet is sent. The type “1” packet includes a series (no longer than 17) of non-used subsequent identifiers. If the series is shorter than 17, the following fields for identifiers are zeroed;Put the obtained data into the EEPROM of R node. Update the field *occupied* using one of the following values:OCCUPIED_NODE_GATEWAY—if the node plays the Gateway role (*RN*&&*GATEWAY*!=0);OCCUPIED_NODE—if the node plays the Master or Replica role (RN!=0);OCCUPIED_EMPTY—for data from the type 1 packet, the RN, NSK, NSK_IV, and NAD fields are zeroed;Repeat the above steps as long as there is a place in the EEPROM memory of the R node or until a type 1 frame with a zeroed N_ID_0_ field (no more data on M node) is received. 

After completing the registration of the first R node in the security domain, the resources of the M node and R node are similar to those shown in Figure 8. The first registered R node in the security domain will play the Gateway role in the domain. After completing the registration of the second and the next R nodes in the security domain, the resources of the M node and R node are similar to those shown in Figure 9.

### 4.3. The Procedure of Restoring Node Data after Powering on

All data stored in the RAM of the sensor node disappears after the power of the node is turned off. An example of the content of the M node resource after turning off the power of the node is shown in Figure 15. The fields of RAM, which have disappeared, are indicated in the figure in light gray.

This procedure is intended to restore data in RAM (N_ID, NAD, MAD, and GAD) based on the records in the node’s TPM NVRAM. The RN field is restored from the node description in the EEPROM. The content of the SQ field is generated randomly. If the data recovery concerns the M node, then restoration of the status for the nodes for which valid data exist in the description of the security domain nodes, which are stored in EEPROM, is also required. Then, the following values are inserted into the fields of status data:*Stat*—*Active*, for the M node;*ActiveNonConfirmed*1* for the other nodes;*Time*—current time in msek;

The content of the node resource after the procedure is similar to that shown in Figure 9. 

### 4.4. Procedure for Refreshing the Node Status

During the procedure described in Section 4.3, the node’s data in the RAM are restored. On the M node, the status fields of the other domain nodes have also been restored, but the current status of these nodes requires confirmation. For this purpose, the procedure for refreshing the node’s status should be used for each node.

The sequence diagram of the procedure for refreshing the node status is shown in Figure 16.

Description of the most critical steps of the procedure are as follows:

**(1) Prepare a request to update the status of the R node**. Do the following:Read the data: N_ID, RN, NSK, and NAD descriptions of the verified R node stored in the EEPROM of the M node;Prepare the ***status_refresh_req*** packet (Figure 17) containing the following data: N_ID, RN, and SQ of the M node. The packet is encrypted using the NSK key of the R node;Send a ***status_refresh******_req*** packet from the M node using NAD as the target address of the R node.

**(3) Verify the correctness of the node description**. Do the following:On the R node, decrypt the received ***status_refresh******_req*** packet and check the correctness of the N_ID and RN fields. If they are not correct, print the message and stop the procedure;On the R node, prepare the ***status_refresh_ans*** packet (Figure 18) and send it to the M node. The SQ from the ***status_refresh******_req*** packet is increased by 1. The packet is encrypted using the NSK key of the R node.

**(5) Acquire node status**: on the M node, update the *Stat* field to the *Active* value in RAM in the description of the verified R node. 

### 4.5. The Procedure for Node Status Confirmation

After the R node data have been restored to RAM, this node initiates the confirmation of its status by sending the ***status_confirm_req*** packet to the MAD address. The purpose of this procedure is to notify the M node of its readiness to work in the domain after the power off/power on cycle and confirm the correctness of the values of the MAD and GAD fields restored in the RAM.

The implied M node (i.e., the one to which the ***status_confirm_req*** packet was sent) will authenticate the querying node if the following conditions are jointly met:(a)The M node is able to decrypt the sent frame using the R node NSK key, which is stored in the EEPROM of the M node in the description of the domain nodes;(b)The content of the NAD field from the frame is consistent with the contents of the NAD field in the description of the domain node;(c)The content of the NTAG field from the frame is consistent with the calculated value based on the data in the M node resources;(d)If the status of the N_ID node in the domain is *Active* or *ActiveNonConfirmed*, then the RN field from the frame must match the RN field in the description of this domain node;(e)If the status of the N_ID node in the domain is *NonActive*, then the RN field from the frame may have *REPLICA* or *REPLICA* + *GATEWAY* content, and the RN field in the description of this domain node must have *REPLICA* content.

If it turns out that the M node being queried no longer fulfills this role, it sends a response that currently this role is being played by another node and gives the current MAD address. Then, the querying node should direct its query to the node with the MAD address.

If the M node does not confirm these data (because, for example, the M node does not work or has crashed or the querying node has not been authenticated), the querying node will not get a response, which consequently, means that this node needs to re-register in the security domain.

The RN field of node R is restored based on the record in its EEPROM and also needs to be confirmed by node M. If, in the meantime, this role has changed (i.e., from *REPLICA + GATEWAY* to *REPLICA*), then the M node sends a reply informing that, currently, the Gateway role is being played by a different node, and additionally, will give the current GAD address and the current contents of the RN field.

The sequence diagram of the procedure for refreshing the node status is shown in Figure 19

The most important steps of the procedure are as follows:

**(1) Prepare a request to confirm the status of the R node**. Do the following:Prepare the ***status_confirm_req*** packet (Figure 20) containing the following data: N_ID, RN, NAD, NTAG, and SQ of R node. The packet is encrypted using the NSK of the R node;Send a ***status_confirm******_req*** packet from the R node using the MAD as the target address of the M node.

**(3) Authenticate R node and update its status**. Do the following:On the M node, decrypt the received ***status_confirm******_req*** packet using the NSK and IV vector of the N_ID node. Authenticate the N_ID node in the domain according to the rules described at the beginning of the section:○If the authentication procedure did not succeed, ignore the packet and terminate the procedure;○If the authentication procedure was successful and the node being requested acted as a Master, set the status of the node N_ID in the domain to *Active* and prepare the package ***status_confirm_ans*** (Figure 21) as follows:N_ID = N_ID gathered from the ***status_confirm_req*** packet;Ans = STATUS_OK;RN = current role of the N_ID node;ADR = GAD—address of the current G node;Time = current time of M node;SQ = the SQ from ***status_confirm_req*** increased by 1.If the authentication procedure was successful, but the node being requested does not have the role of Master, then prepare the ***status_confirm_ans*** package as follows:N_ID = N_ID gathered from ***status_confirm_req*** packet;Ans = STATUS_NEW_MAD—the other node is playing the MASTER role in the domain;ADR = MAD—address of the current M node;SQ = SQ from ***status_confirm_req*** increased by 1;The other fields are zeroed.Encrypt the ***status_confirm_ans*** packet using the NSK of the N_ID node and send it to the R node.

**(5) Acquire node status** confirmation—on the R node, depending on the content of the *Ans* field, do the following:If *Ans* = *STATUS_OK*, in the RAM of the node, update the RN, GAD, Time, and SQ field values. Update the GAD field in the NVRAM of the node;If *Ans* = *STATUS_NEW_MAD* in the RAM and the NVRAM of the node, update the value of the MAD field. Perform the procedure for node status confirmation again with the new MAD address.

## 5. Results and Discussion

### 5.1. Security Domain Demonstrator

The presented concept of a secure domain for sensor nodes has been experimentally tested. A domain demonstrator covering several sensor nodes was prepared for the needs of the research. Each of these sensor nodes (Figure 22) consisted of the following components: an Arduino Mega 2560 R3 microcontroller, a TPM (Used module meets the requirements described in Security Policy for Atmel TPM [21] as well as the strength requirements of FIPS 140-2, Level 2 [22]) v.1.2 installed on Cryptotronix CryptoShield, Dragino LoRa Shield 868 MHz v1.3, and a wireless communication module, XBee. The role of the measuring sensor, which was the source of sensor data in the domain, was performed by the ultrasonic distance sensor. 

The designed sensor nodes were used to examine all the procedures of the domain deployment phase. The result of these procedures was the creation of a secure domain including several sensor nodes, one of which served as a Master, one as the Gateway, and the other sensor nodes as Replicas. Selected elements of a functioning domain during the research are shown in (Figure 23).

### 5.2. Analysis of Wireless Bandwidth Consumption

When designing solutions for the IoT, the small resources of IoT nodes are a challenge. In particular, the problem relates to the limited energy resources of the power sources that can be used. The analysis of the efficiency of energy consumption and its saving is complicated and very dependent on the specific conditions of the operation of the IoT network nodes. Undoubtedly, elements such as the type of wireless interface used, the expected reach of a single node and the entire network, the method of data exchange between nodes, the volume of data necessary for transmission between nodes, among others, have significant influences on energy consumption. The framework of this study did not allow for a detailed analysis of the problems related to energy consumption. To illustrate the expected costs when using the proposed solution to protect the functioning of the domain of the sensor nodes, the following sections present an analysis of the bandwidth consumption for the most critical procedures of the phase of creating a secure domain of sensor nodes. 

#### 5.2.1. Registering Nodes in a Domain

The procedure of registering one node in the domain requires the exchange of packet sequences between the node being registered and the M node. The list of names of these packages, their required size, direction, and the number of transfers are shown in Table 1. 

An XBee interface was used to exchange data between the domain nodes. Due to the specificity of this interface, the size of a single frame sent by this interface cannot be larger than 64 bytes. For this reason, the packages described in Table 1 must be fragmented. For example, the contents of one ***transfer_key_req*** packet containing 272 bytes will be sent in six XBee frames, of which five will have a maximum size of 64 bytes (full frame), and the last will be 48 bytes. In each full frame, 16 bytes of the XBee header and 48 bytes of the ***transfer_key_req*** packet will be sent, and in the previous frame, the payload will cover the remaining 32 bytes of this packet. Packet fragmentation increases the required volume of data to be transmitted over the link—the numbers of XBee frames and bytes to be transmitted in these frames are given in the last two columns of Table 1.

In the sequence of packages necessary to register one node in the domain, only the packets used to send descriptions of domain nodes (i.e., ***back_up_node_req*** and ***back_up_node_ans***) are used repeatedly, and the others are used only once. It is worth noting that the number of these packets is proportional to the number of already registered nodes in the domain. This is because the description of one registered node requires the whole of one ***back_up_node_ans*** packet (type (0)), and in the type (1) of the ***back_up_node_ans*** packet, it is possible to send up to seventeen descriptions of non-registered nodes. The dependence of the number of required XBee frames in the procedure of registering one node from the number of nodes already registered in the domain is given in Figure 24, and the dependence of the number of bytes to be transmitted in these frames is shown in Figure 25.

The data volume to be transmitted for a domain with several dozen nodes is quite large and for N = 60, it is 10,356 bytes. The time needed to register the tenth node in the sensor domain was experimentally tested. This procedure required 1168 bytes to be sent towards R → M and 1892 bytes towards M → R. The registration time for this node was a few seconds—the XBee interface was operating at 57,600 bps. 

#### 5.2.2. Refreshing the Status of Domain Nodes after Powering on the M Node

The procedure for updating the domain node status was initiated after powering on the M node. Table 2 presents the transmission requirements for the refresh status procedure for one registered node in the domain. 

This procedure must be performed for each registered node in the domain. The number of sent bytes required to refresh the status of all registered nodes in the domain will be the product of the values given in the last column of Table 2 and the number of registered nodes in the domain.

#### 5.2.3. Confirmation of the R Node Status in the Domain Description after Power up

The procedure to confirm the status of node R must be done once after turning on the power supply of this node. Table 3 presents the transmission requirements for the procedure of confirming the status of the R node.

If, from the moment of the last activity of the enabled R node in the domain, there were changes, and if, currently, the Master role is performed by another node, the MAD address of the node that acts presently as Master will be sent in the ***status_confirm_ans*** frame in the ADR field. In this case, the enabled R node must retry the procedure using the newly received MAD address.

The study focused on the assessment of the feasibility of the proposed solution and its scalability. The main conclusion resulting from the obtained results (presented in the Table 1, Table 2 and Table 3, and Figure 24, Figure 25 is the linear dependence of the domain deployment time on the number of its nodes. The same conclusion can be transferred to energy consumption. Energy consumption for domain preparation also increases linearly with the increase in the number of domain nodes. Thus, it showed that the proposed solution is scalable.

### 5.3. Security Evaluation

A comprehensive security solution for the critical application of IoT should include three basic features: preventative, detective, and reactive measures [4]. Our framework for constructing a secure domain of sensor nodes mainly focused on the first type of measures. This indicates that it is aimed at prevention or at the very least, at making some common attacks much more difficult. However, this does not mean that the application of detective and reactive measures has not been foreseen in the proposed solution. 

We propose some diagnostic procedures built upon system level diagnosis where the sensor nodes of the domain are capable of performing assigned tests among themselves. After performing all mutual tests between nodes, based on the test results obtained, the nodes that do not work correctly can be identified (with certain restrictions). By using the appropriately designated structure of mutual node testing, the domain’s ability to detect or locate faulty or (misbehaving) nodes can be obtained.

After identifying the misbehaving nodes, a domain reconfiguration is carried out, which, in the case of failure/malfunction of the nodes that perform the Replica role (R nodes), excludes them from the domain. If a malfunctioning M or G node is identified, the procedure for selecting a new node to perform these roles is started. All diagnostic and reconfiguration procedures are protected cryptographically using the NDK (different symmetric keys for messages encryption ) and are limited to authenticated nodes only.

The main difficulty in securing wireless IoT networks is the fact that sensor nodes could be unattended. This leads to the threads that are not viable in another type of network. Let us consider common real-life threads and attacks for wireless sensor networks, such as malicious sensor node injection (node replication attack), sensor impersonation (imitation), attacks on the information in transit, denial-of-service attacks (DOS attacks), and routing attacks [4]. 

#### 5.3.1. Node Replication Attack

An attack involving the addition of a cloned sensor node to the network is quite simple in an unsecured network. In the case of our solution, it is not possible, because each TPM has a unique asymmetrical EK that can not be deleted or regenerated. This key is at the top of the key hierarchy that forms the trust structure. There is also a DK in this structure. Creating a domain-acceptable trust structure on another specimen of the sensor node requires the DK to be obtained. One of the methods of obtaining this key may be the execution of the pre-preparation procedure of the node, but this procedure is performed only in a safe space and requires direct connection of the prepared node to the B node. It should be recognized that such an attempt is unlikely to be carried out by unauthorized persons. Another method might be to obtain a DK key from a functioning node in the domain. However, this approach requires interference in the software and configuration of the sensor node. These types of activities are detected by TPM module mechanisms based on PCRs. If a node’s integrity violation is detected during its regular operation, the sensor node sends an appropriate message to the remaining nodes of the domain and is then considered untrusted. If the integrity violation is detected when it is launched, the node’s software is blocked, and re-use of such a sensor node is possible after the node”s pre-preparation procedure is completed, and then the node is registered in the domain.

#### 5.3.2. Sensor Impersonation

Sensor impersonation is not possible. Each sensor node has a fixed identifier (N_ID), an NSK key, and a special tag (NTAG). All of these parameters are determined during the pre-preparation procedure of the node and used during the registration procedure in the domain. The NTAG tag is important. Determining the value of this tag requires knowledge of the domain name, the node identifier, and other fields that are known only to node B and node M.

#### 5.3.3. Attack on Information in Transit

The transmitted information is vulnerable to eavesdropping, modification, injection, interruption, and traffic analysis. Eavesdropping, modification, injection, and interruption are protected in our solution by assuring authentication, confidentiality, and integrity techniques, and also by using replay protection, which is based on SQs. The sequential number is sent in each frame, and next time it is used, it is increased by 1. The sequence number is always placed in the encrypted part of the frame. Such an approach, admittedly, extends the detection time of the repeated frame, but hides the current value of the number against the attacker.

#### 5.3.4. Denial of Service

A DOS attack at the physical level is beyond the scope of protection of the secure domain mechanisms, and protection against this type of threat requires completely different means (e.g., recent developments have resulted in a prototype DM antenna system that applies physical layer encryption on top of existing radio–communication interfaces [23]). Denial of service attacks can also be performed at the data link and network layers. Network layer DOS attacks involve the routing protocol. 

#### 5.3.5. Routing Attacks

There are a number of attacks that target the routing protocol of WSNs. Since all such attacks, like Selective Forwarding, Sybil Attack, Wormhole Attack, Sinkhole Attack, or False Routing Information, are related to the placement of a malicious node inside the network or the manipulation of an existing domain sensor node, the possibilities of performing such attacks in the secure domain are very limited. Continuous monitoring of software integrity and hardware configuration of the sensor node using PCRs of TPM practically eliminates the possibility of installing malicious software.

#### 5.3.6. Botnet Activities

One of the significant threats to the IoT network is botnets. Internet of Things botnets have received only sporadic attention from researchers. The bot is malware that infects IoT devices and aims to propagate the infection to misconfigured devices and to attack a target server after it receives the corresponding command from the person controlling the bot, or the botmaster. An example of a bot is Mirai malware [24]. Mirai primarily spreads by first infecting IoT devices such as webcams, DVRs, and routers that run some version of the software (BusyBox). It then deduces the administrative credentials of other IoT devices by means of brute force. In 2016, a spectacular distributed denial-of-service attack (DDoS attack) using Mirai malware (with at peak 1.1 Tbps) was performed, which targeted the French webhost and cloud service provider On Vous Héberge (OVH) [24]. The solutions proposed within the framework provide resistance to bot injection because they prevent the sensor nodes of the secure domain from software manipulation by using PCRs of the TPM.

### 5.4. Performance Evaluation

The demonstrator’s preparation aimed to prove the usefulness and correctness of the concept of a security domain of sensor nodes and to assess the system’s security. The evaluation of performance in terms of detailed computational requirements was not the primary goal of this work. Assuming the TPM performed all activities related to generating cryptographic keys, encrypting and decrypting data, and determining the value of the hash function, we made some related observations. The efficiency of the TPM module when performing these activities varied. For example, generating an asymmetric RSA-2048 key takes the most time and could take up to ninety seconds. Fortunately, this operation is only carried out in the phase of pre-preparation of the node, in which there is no time pressure. The remaining activities require from a few to several dozen milliseconds. The time of reading data from the NVRAM of the TPM, which is about 46 ms, is significant.


**Memory utilizing**


The basis for the construction of each sensor node is the Arduino Mega 2560 R3 module, which uses the AVR ATmega 2560 microcontroller. This microcontroller has 256 kB Flash, 4 kB EEPROM, and 8 kB RAM. Flash memory is used in about 60%. The EEPROM size limits to sixty the number of sensor nodes that can be registered in the domain. The size of RAM was a very big limitation for the implementation of the solution. Such a small memory forced the allocation of most data structures in dynamic memory and releasing it immediately after use. This approach, in particular, concerned data structures for cryptographic keys. It is worth realizing that the data structure containing the private and public part of one asymmetric RSA2048 key requires about 800 bytes of RAM—it is about 10% of accessible RAM. The use of dynamic memory allocation extends the initialization time of data structures, also because the cryptographic keys that are stored in the NVRAM of the TPM are necessary to initialize these structures. However, this approach allows for more effective use of RAM.


**Delay of the cryptographic protection procedures**


When creating software for the sensor node, it was assumed that the cryptographic keys would be stored in protected TPM resources, but in the explicit form in RAM they will only reside during the execution of procedures that require these keys. From the point of view of security, this approach is correct, but it extends the duration of the procedures utilizing the cryptographic keys. In order to check what delay causes cryptographic protection of the sent frame, an experiment was performed in which the same data collected from the distance sensor was sent to the G node first without cryptographic protection and then using the Advanced Encryption Standard (AES) algorithm. The data were sent in the sensor_packet_req frame and the confirmation of receipt of this data in the sensor_packet_ans frame. The results of the experiment are given in Table 4.

The time needed to handle an encrypted frame is about 83 ms longer than serving an unencrypted frame. It is worth noting here that the reading time of the encryption key from NVRAM of the TPM is about 46 ms. Of course, it is possible to resign from downloading of the key from the NVRAM of the TPM whenever needed and keep this key in RAM, but such an approach is less secure and further increases the RAM deficit.

## 6. Conclusions

The usefulness of the presented framework was experimentally confirmed. The proposed framework could be easily extended by developing procedures for building trust between cooperating domains (clusters). It will enable secure data exchange between the security domains of sensor nodes. For military applications, in addition to security, one of the basic parameters of the applicability of the solution is the time of system preparation and its scalability. The experiments confirmed that the time of secure domain deployment is fully acceptable. The linear dependence of the number of operations performed, depending on the number of nodes creating the domain, was determined experimentally. 

Significant activities of the first stage of preparing a secure sensor node domain are described in detail in Reference [15]. The procedures presented in this study relate to the activities of the second stage of creating a secure domain of sensor nodes including, in particular, initiation of the first sensor node of the domain, registration of subsequent sensor nodes in the domain, and refreshment and confirmation of the status of the domain nodes. Other activities that are intended to be performed during the regular operation of the created domain are not described in this paper.

In the presented procedures, particular attention was paid to the security of procedures—only sensor nodes prepared in the first stage had the possibility of effectively implementing the second-phase procedures. During the execution of each of the second-stage procedure, the XBee wireless link was used for data exchange. The asymmetrical cryptography (RSA-2018) was only used in the first step of the registration procedure of the sensor node in the domain to protect the data transmitted. In all other data protection activities (both those stored in the node’s resources and those transmitted between domain nodes), AES symmetric cryptography was used in CBC (Cipher Block Chaining) mode. All described procedures were supported by TPM.

During the registration procedure of the node in the domain, a special NTAG tag was verified, which was generated for each sensor node during the preparation of this node in the first stage. The use of this tag eliminated the possibility of registration occurring in the domain of the node in which unauthorized impact in the resources of the sensor node occurred before this node was registered in the domain.

The procedures necessary to present the operation of the secure domain of sensor nodes were implemented. For this purpose, a domain demonstrator comprising four mobile sensor nodes was prepared, one of which served as a Master, another node acted as a Gateway, and the others worked as Replicas. During the experiments, data generated by nodes of a secure domain were collected by the G node and sent to the recipient via the prepared MQTT broker. Selected results of performed experiments are presented in Reference [15].

Our future work will include the implementation of procedures for detecting damaged/misbehaving sensor nodes of the domain. Another crucial issue is the development of procedures for selecting new nodes to perform the Master and Gateway roles in case of damage to a node that has performed such roles and procedures for protecting sensor nodes from unauthorized influences on its hardware configuration or its data resources. The second direction of our future study will be to modify solutions using newer TPM implementations in version 2.0 and cryptographic algorithms based on elliptical curves.

## Figures and Tables

**Figure 1 sensors-19-02797-f001:**
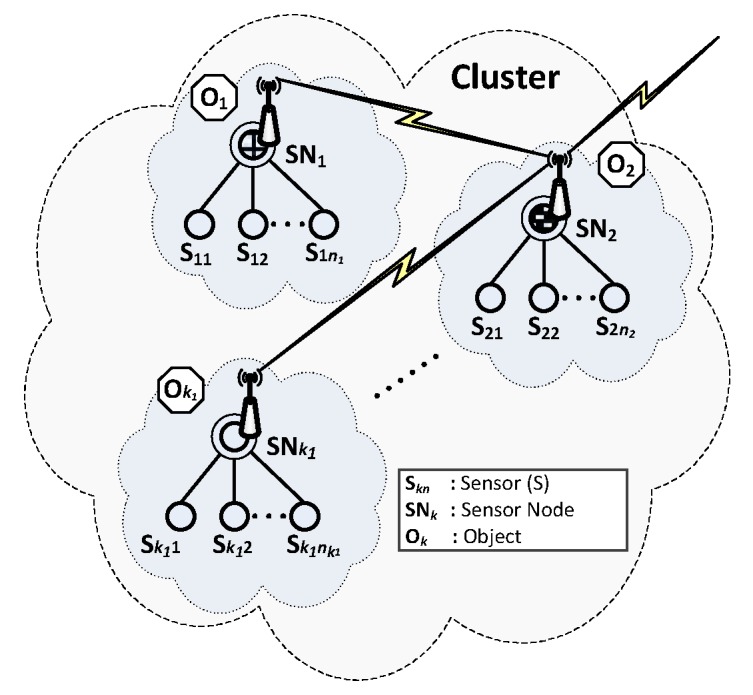
Structure of a sensor node network cluster.

**Figure 2 sensors-19-02797-f002:**
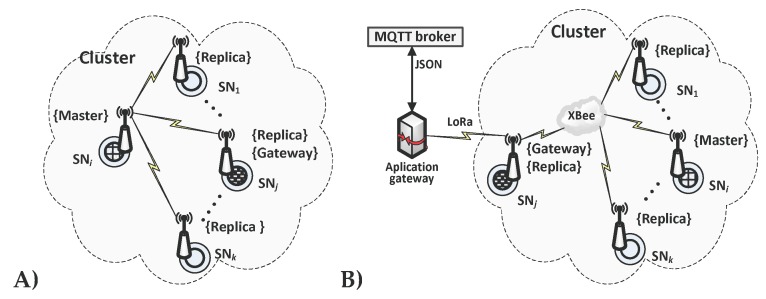
The logical topology of the network in the security domain (**A**) and network in the domain of the transmission of data obtained from sensor nodes (**B**).

**Figure 3 sensors-19-02797-f003:**
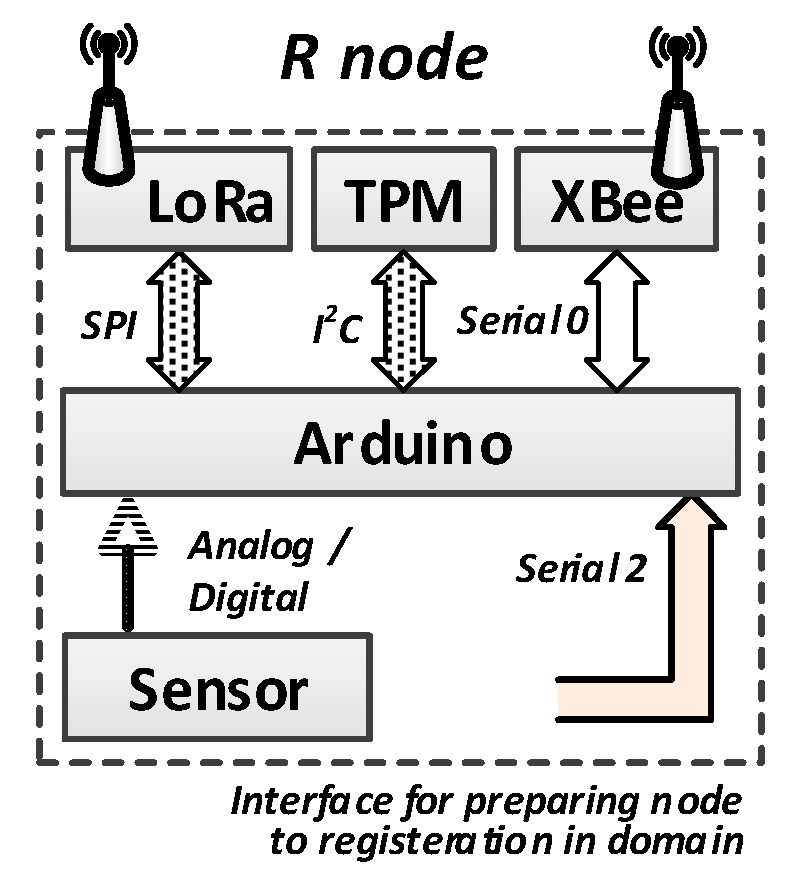
Block diagram of the sensor node. Arduino: single-board microcontroller; I^2^C: Inter-Integrated Circuit; LoRa: interface for long range wireless trenssmissions with low power consumption; SPI-Serial Peripheral Interface; TPM: Trusted Platform Module; XBee: interface for low range wireless trenssmissions;

**Figure 4 sensors-19-02797-f004:**
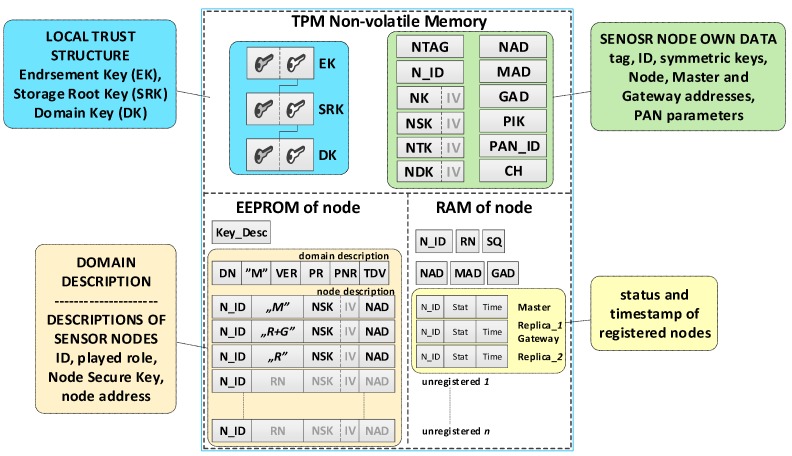
An example of data stored on the M node and R node. CH: Operating channel in a wireless network; DK: Domain Key; EK: Endorsement Key; GAD: Gateway Node address; MAD: Master Node address; NAD: Node address; NDK: Node Diagnostic Key; NK: Node Key; NSK: Node Secure Key; NTAG: Node tag; NTK: Node Transmission Key; PAN_ID: Identifier of personal area network; PIK: PAN Interface Key; SQ: Node sequential number; SRK: Storage Root Key; TPM: Trusted Platform Module.

**Figure 5 sensors-19-02797-f005:**
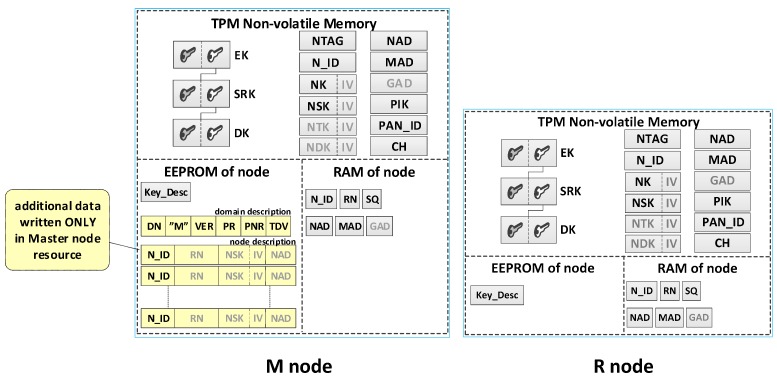
Differences in the content of saved data on the M node and R node after preparing the procedure.

**Figure 6 sensors-19-02797-f006:**
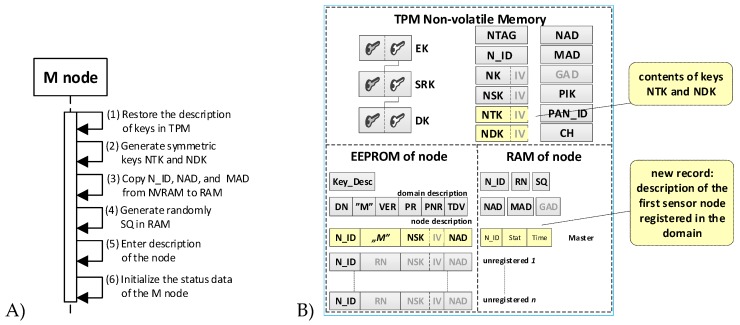
The sequence diagram for the M node initialization procedure (**A**) and data stored on the M node (**B**) after the end of the initialization procedure.

**Figure 7 sensors-19-02797-f007:**
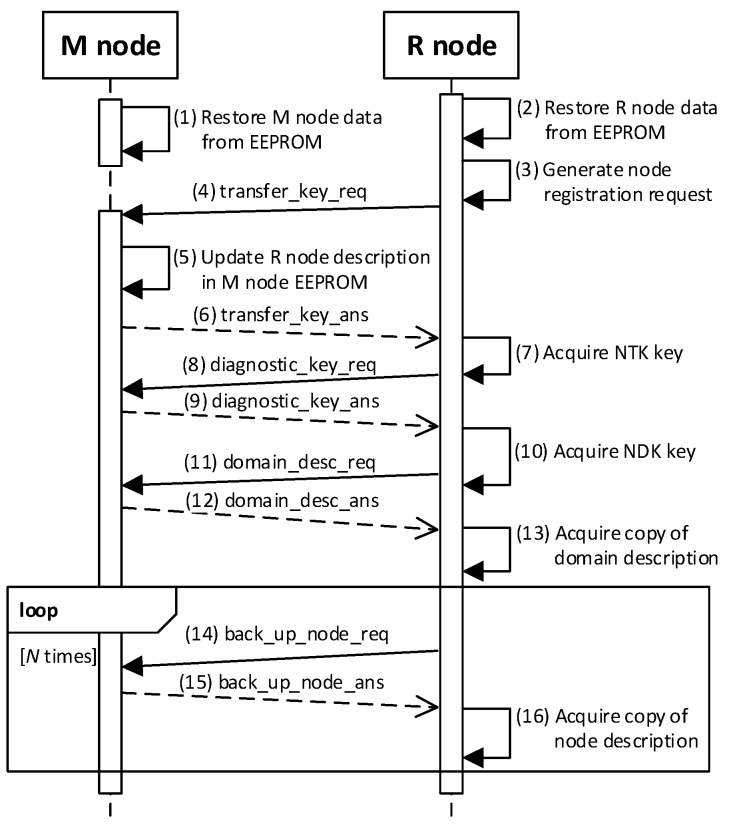
The sequence diagram for the R node registration procedure.

**Figure 8 sensors-19-02797-f008:**
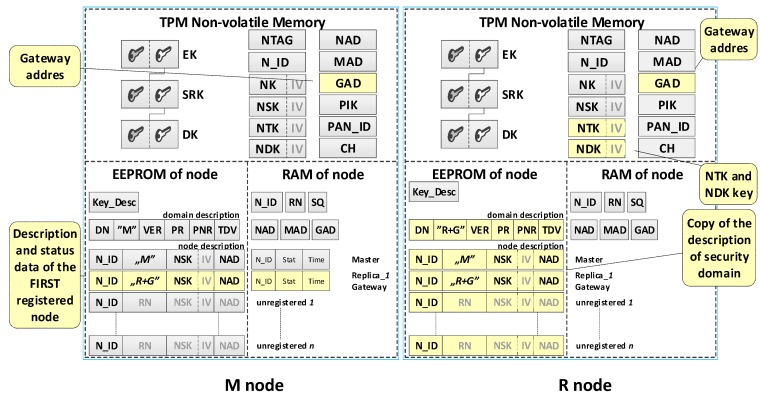
The data updated on the M node and R node (highlighted in yellow) at the end of the procedure of registering the FIRST R node in the security domain.

**Figure 9 sensors-19-02797-f009:**
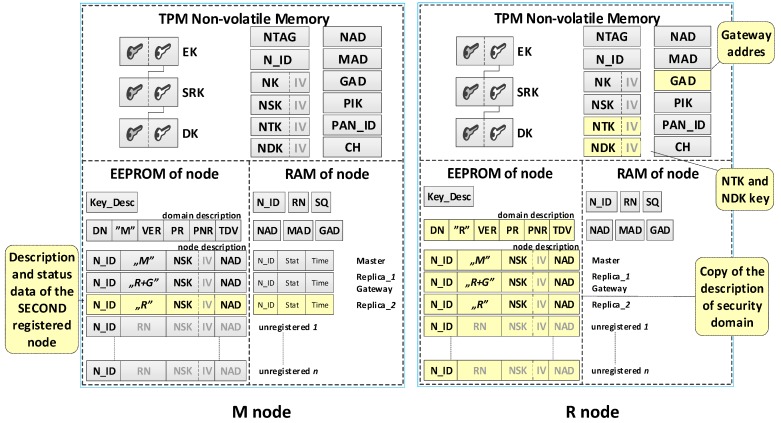
The data updated on the M node and R node (highlighted in yellow) at the end of the procedure of registering the SECOND R node in the security domain.

**Figure 10 sensors-19-02797-f010:**
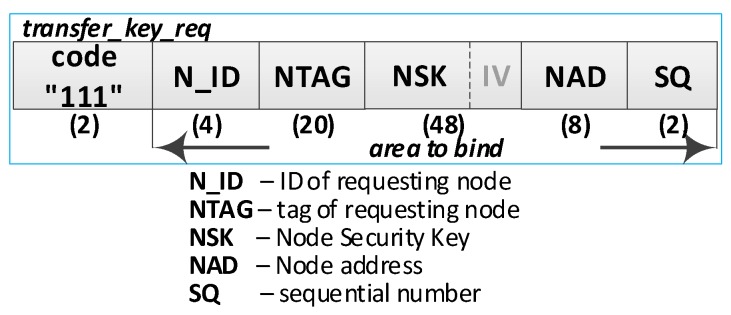
The node registration request frame.

**Figure 11 sensors-19-02797-f011:**
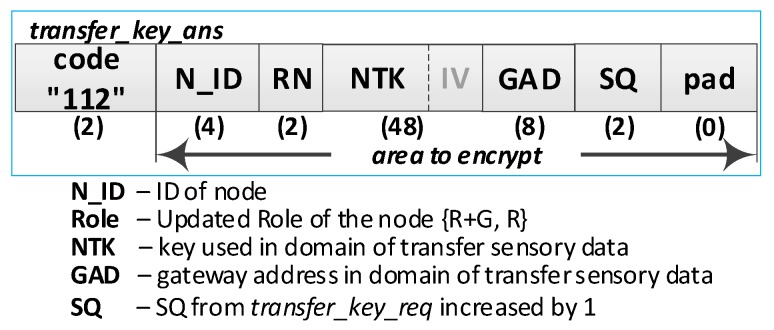
The confirmation frame for node registration. SQ: sequence number.

**Figure 12 sensors-19-02797-f012:**
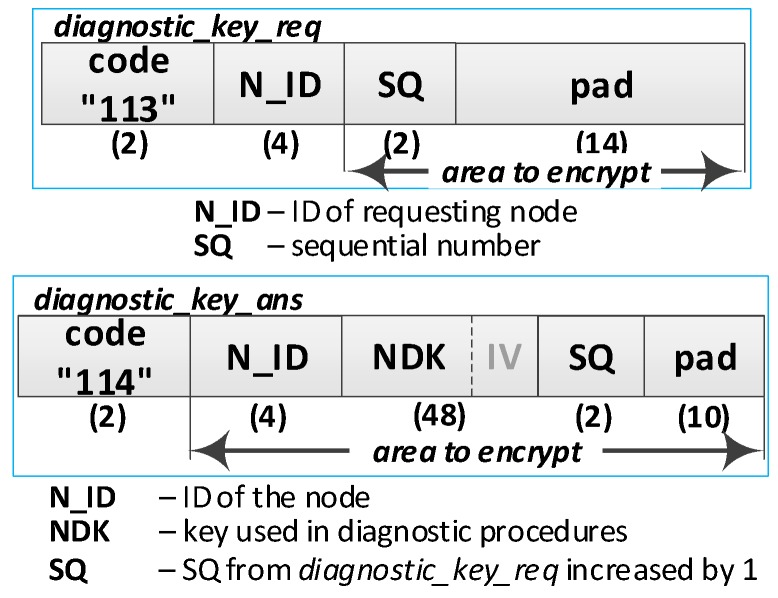
The frames used by R node to acquire the NDK.

**Figure 13 sensors-19-02797-f013:**
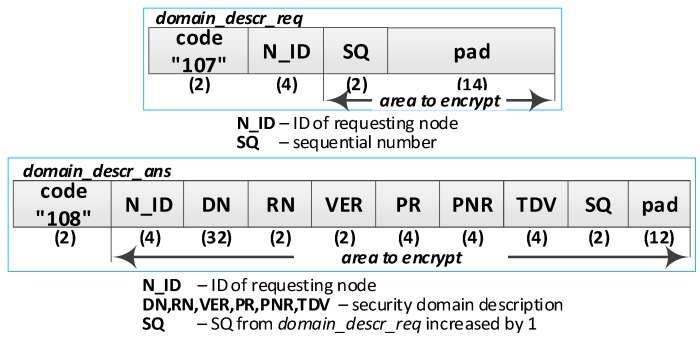
The frames used by R node to acquire a copy of the domain description.

**Figure 14 sensors-19-02797-f014:**
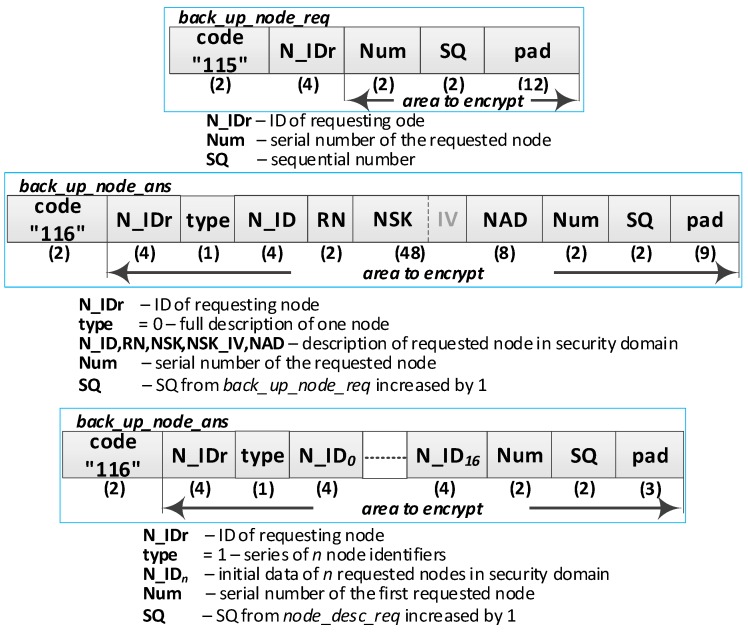
The frames used by the R node to acquire a copy of the description of the domain nodes.

**Figure 15 sensors-19-02797-f015:**
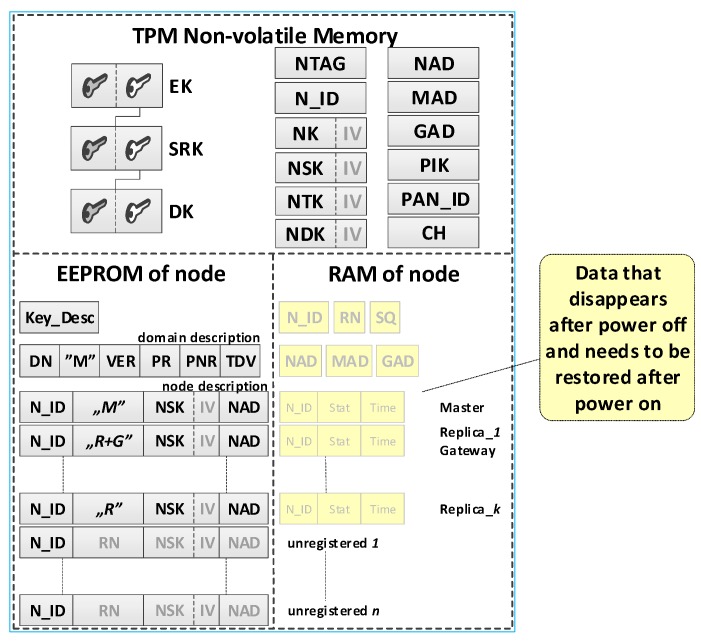
The data stored on the M node after turning off the power of the node.

**Figure 16 sensors-19-02797-f016:**
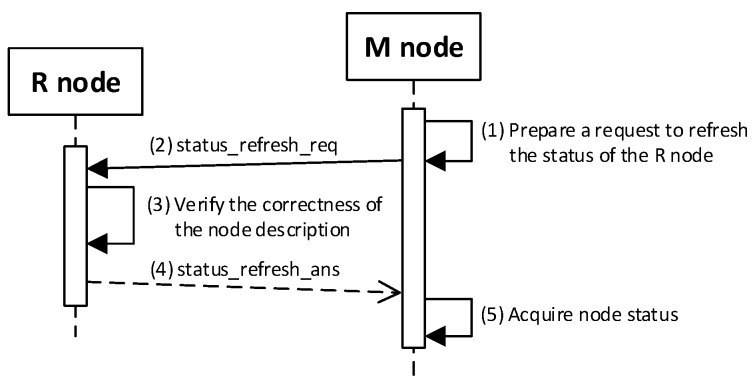
The sequence diagram for the procedure for refreshing the node status.

**Figure 17 sensors-19-02797-f017:**
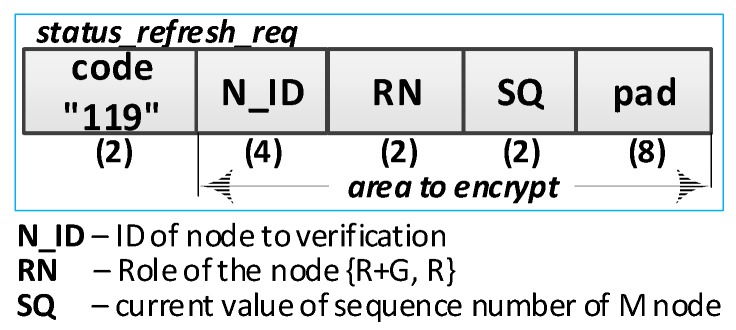
Request frame for refreshing node status.

**Figure 18 sensors-19-02797-f018:**
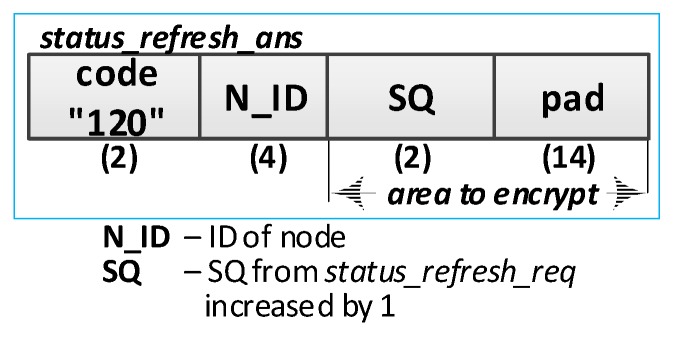
Response frame for node status refresh request.

**Figure 19 sensors-19-02797-f019:**
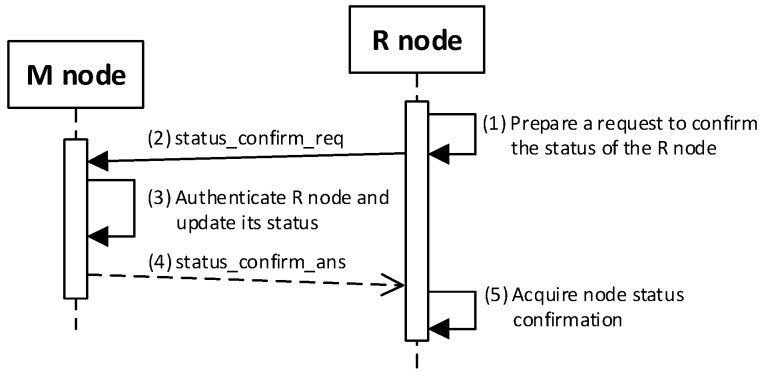
The sequence diagram for the procedure for node status confirmation.

**Figure 20 sensors-19-02797-f020:**
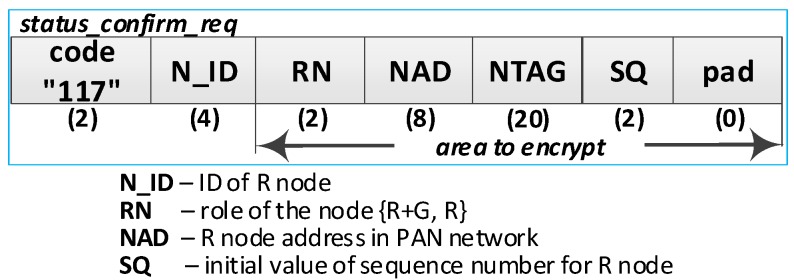
The confirm node status request frame.

**Figure 21 sensors-19-02797-f021:**
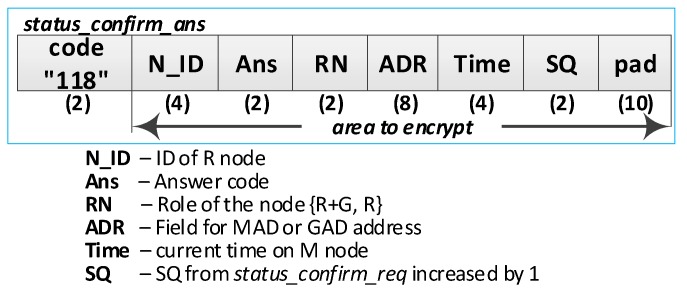
The confirmed node status answer frame.

**Figure 22 sensors-19-02797-f022:**
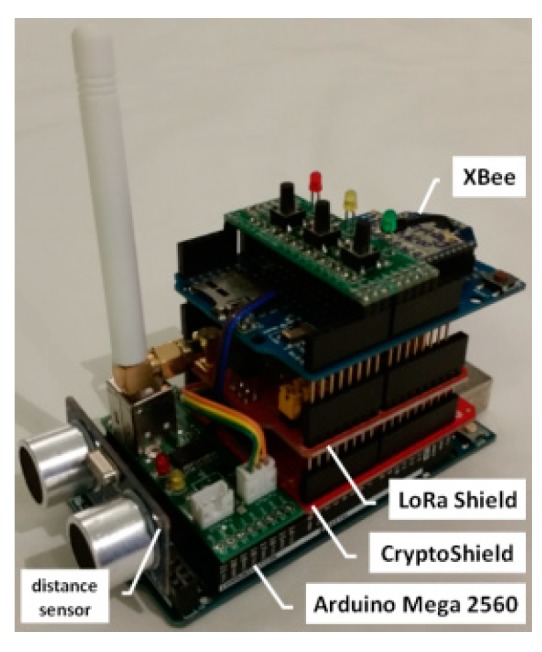
View of the sensor node demonstrator.

**Figure 23 sensors-19-02797-f023:**
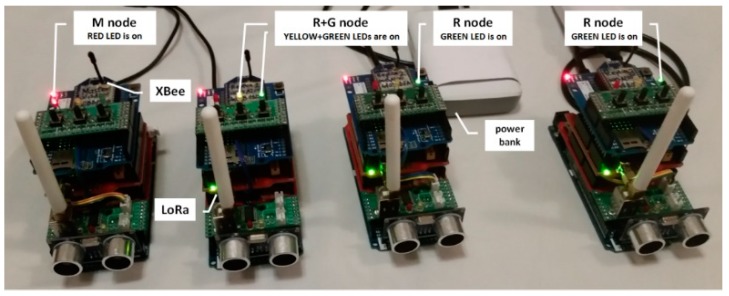
View of the demonstrator during the transmission of sensor data to the G node.

**Figure 24 sensors-19-02797-f024:**
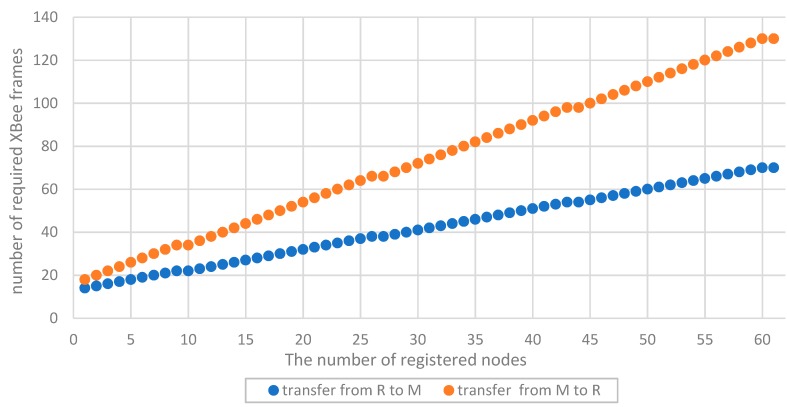
Number of nodes to register versus the number of required XBee frames to be transferred during the procedure for registering the *N*th node in a domain.

**Figure 25 sensors-19-02797-f025:**
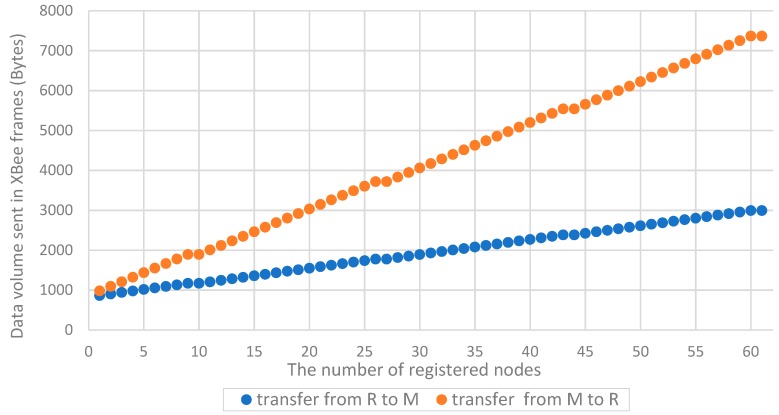
The number of nodes to register versus the data volume needed to send in XBee frames during the procedure to register the *N*th node in a domain.

**Table 1 sensors-19-02797-t001:** Transmission requirements for the procedure for registering the first node in the domain.

Packet Name	Packet Length (B)	Direction	Required Number of Packets	Number of Bytes in Packets (B)	Number of Xbee Frames	Number of Bytes in Xbee Frames (B)
transfer_key_req	272	R → M	1	272	6	560
transfer_key_ans	66	M → R	1	66	2	98
diagnostic_key_req	22	R → M	1	22	1	38
diagnostic_key_ans	66	M → R	1	66	2	98
domain_desc_req	22	R → M	1	22	1	38
domain_desc_ans	68	M → R	1	68	2	100
back_up_node_req	22	R → M	6	132	6	228
back_up_node_ans (0)	82	M → R	1	82	2	114
back_up_node_ans (1)	82	M → R	5	410	10	570
	Total		18	1140	32	1844

**Table 2 sensors-19-02797-t002:** Transmission requirements for the procedure for refreshing the status of one node.

Packet Name	Packet Length (B)	Direction	Required Number of Packages	Number of Bytes in Packets (B)	Number of Xbee Frames	Number of Bytes in Xbee Frames (B)
status_refresh_req	18	M → R	1	18	1	34
status_refresh_ans	22	R → M	1	22	1	38
	Total		2	40	2	72

**Table 3 sensors-19-02797-t003:** Transmission requirements for the procedure confirming the status of one node.

Packet Name	Packet Length (B)	Direction	Required Number of Packages	Number of Bytes in Packets (B)	Number of Xbee Frames	Number of Bytes in Xbee Frames (B)
status_confirm_req	38	R → M	1	38	1	54
status_confirm_ans	34	M → R	1	34	1	50
	Total		2	72	2	104

**Table 4 sensors-19-02797-t004:** Transfer time of sensor data acquired by the Gateway node [17].

Value of Distance (mm)	Protect	Packet Length (b)	sensor_packet_req (ms)	Transfer (ms)	sensor_packet_ans (ms)
56	NO	22	7.13	12.91	0.28
56	AES	22	87.29	12.94	86.87

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
