# Peer review of "A Framework for Constructing a Secure Domain of Sensor Nodes"

_sensors, 2019, doi:10.3390/s19122797_

Round 1
Reviewer 1 Report
The paper deals with a really interesting topic (security in WSN) and it adopts a very practical approach. I found the work done by the authors very good and the paper quite clear.
To help the reader and to improve the quality of the manuscript I suggest to modify/consider the following aspects:
1) find a way to report from Fig.4 to Fig.15 in another fashion to avoid boring the reader and move its attention from the content
2) add a discussion on the impact of the proposed framework on the well known energy aspects. For example consider the aspect of sleep scheduling and the way the proposed apprach can deal with them and benefit from sleep scadeling (https://doi.org/10.1016/j.jnca.2016.12.022) and relation with energy consuption at large (https://doi.org/10.1109/SAS.2009.4801782, https://doi.org/10.3390/s17091978, https://doi.org/10.1109/ISCC.2003.1214142). Please consider the reported ref, discuss them and find other relevant ref for this discussion.
3) improve the qualitu of figure 24 and 25
4) (qualitatively) discuss the results, table and figures - again - with respect to energy aspects
This is one the best paper I have reviewed for Sensors till now.
Author Response
1. The arrangement of drawings 4, 5, 6, 13, 14 and 15 has been changed by adding additional descriptions and marking with yellow colour the elements relevant from the point of view of the content presented. Below the drawings, additional descriptions were added to content of the paper. The description of the results of the node registration procedure in the domain has been extended (l.519 - l.532).
2. Aspects related to energy
We are aware of the importance of effective energy management in the normal phase of the system operation. However, as part of this work, we focused mainly on the phase of preparing the domain for work. As the procedures within this phase will usually be carried out in a controlled environment, energy savings are less important here. For most military applications, it is important to maintain operational capabilities for the duration of the mission (survivability).
For military applications, the overriding goal is to provide the maximum level of security. The increased energy consumption in our solution results mainly from the use of TPMs and the use of several cryptographic keys to build a local and domain trust structure, as well as from the use of redundancy in data structures describing domain trust relationships. To the best of our knowledge, we did not find similar solutions, so we can’t compare our solution with others. According to the measurements made in our implementation, the use of TPM in the normal phase of domain work increases the energy consumption of a single node by about 10% in average.
However, we want to reduce energy consumption in the phase of normal domain operation. As we indicated in the Conclusion section, we plan to soon transfer this solution to the TPM 2.0 platform. This will give us new opportunities to save energy by using the special TPM 2.0 sleep mode. It is expected that in a normal environment, the device could be in an internal sleep state for approximately 90% of the operating time of the platform. Assuming TPM 2.0 is operating in the internal sleep state with active clock we have the possibility to reduce energy consumption by the TPM module to approximately 40% [Estimated on the base of: Infineon SLB 9665 TPM2.0 Trusted Platform Module, Data Sheet Revision 1.0, 2015-10-27].
3. The quality of the graphs in figures 24 and 25 has been improved
4. Discussion of results in the context of energy
Obtained results during experimental research are mainly related to the assessment of the feasibility of the domain preparation phase and the scalability of the proposed solution. The main conclusion resulting from the obtained results (presented in the Tables 1,2,3 and the Figures 24-25) is the linear dependence of the domain preparation time on the number of its nodes. The same conclusion can be transferred to energy consumption. Energy consumption for domain preparation also increases linearly with the increase in the number of domain nodes.
Reviewer 2 Report
The authors propose a framework to provide security measures with the purpose to build a “secure domain of sensor nodes”. Their proposal is based on a Trusted Platform Module (TPM) module located inside the sensor nodes (IoT devices).
The work is based on (essentially constitutes an extension of) previous works of the same group of authors, i.e., [8], [9] & [R1]. Although the authors do not explicitly mention this fact, they have already published several building blocks of the framework and conducted experiments in these 3 works, like the concept of the secure domain, various phases for building the secure domain, and others. In addition, the majority of the figures and the implementation of the testbed are copied, as it is, from these publications. Therefore, the authors should state clearly what are the differences between the current work compared with their previous works and justify its novelty and its actual contributions.
From the reviewer’s perspective, it seems that the current work is a detailed explanation of the implementation of the proposed framework, which does not merit a publication in Sensor Journal.
To strengthen their manuscript, the authors should consider and demonstrate examples of how the proposed framework is able to protect from real life attacks, such as malicious sensor node injection, sensor impersonation, or legacy network attacks. In addition, it will be useful to consider and explain how the framework will protect the sensors from entangling in botnets activities, as it happened in the case of Mirai botnet [R2].
The introduction section starts with a sentence stating that they do not exist security measure for the case of IoT in critical areas, like military, etc., applications. However, in very recent years, a lot of effort has been placed in this direction. For example, these works [R3, R4] specifically deal with the security of military IoT. So, I will recommend the authors to revise their introduction statement accordingly.
At the end of the Introduction section, the organization/structure of the paper is missing.
In their proposal, the authors neglect to describe adequately the structure and functionality of the TPM. Or at least, it is not evident throughout the technical details of the implementation. In overall, I will suggest that before the detailed description of the message exchanges, it will be useful to provide a high level overview of the framework and its procedures. In this abstract description, they should also explain which security services each procedure aim to achieve and which security primitives utilize. In the current presentation, the essential parts (namely, the provided security services) are lost inside the technical details.
Furthermore, it is not clear how the symmetric key distribution is taken place in a secure manner.
An evaluation of performance in terms of computational requirements is missing. The authors only provide the communication cost in terms of how many additional bytes/packets the proposed procedures introduce.
The reference section contains only 9 citations (excluding self-citations). Previous works in the area are not adequately recognized. Therefore, they should strengthen the related work section with the suggested works [R2-R9], taken from various concepts that they deal and combine (namely, key management in IoT environment, secure domain of sensor nodes, secure data transmission in IoT, etc.), and other recent related works that they survey.
Some typos:
Fig. 4: indyvidual --> individual
l. 368: round brackets --> parenthesis
References:
[R1] Furtak, Janusz, and Jan Chudzikiewicz. "Secure Transmission in Wireless Sensors’ Domain Supported by the TPM." International Conference on Innovative Network Systems and Applications. Springer, Cham, 2015.
[R2] Kolias, Constantinos, et al. "DDoS in the IoT: Mirai and other botnets." Computer 50.7 (2017): 80-84.
[R3] Agrawal, Sarita, and Manik Lal Das. "Internet of Things—A paradigm shift of future Internet applications." 2011 Nirma University International Conference on Engineering. IEEE, 2011.
[R4] Benabdessalem, Raja, Mohamed Hamdi, and Tai-Hoon Kim. "A survey on security models, techniques, and tools for the internet of things." 2014 7th International Conference on Advanced Software Engineering and Its Applications. IEEE, 2014.
[R5] Kodali, Ravi Kishore, Sushant Chougule, and Ashok Agarwal. "Key management technique for heterogeneous wireless sensor networks." IEEE 2013 Tencon-Spring. IEEE, 2013.
[R6] Badawy, Ahmed, et al. "Unleashing the secure potential of the wireless physical layer: Secret key generation methods." Physical Communication 19 (2016): 1-10.
[R7] Lu, Huang, Jie Li, and Mohsen Guizani. "Secure and efficient data transmission for cluster-based wireless sensor networks." IEEE transactions on parallel and distributed systems 25.3 (2014): 750-761.
[R8] Tang, Di, et al. "Cost-aware secure routing (CASER) protocol design for wireless sensor networks." IEEE Transactions on Parallel and Distributed Systems 26.4 (2015): 960-973.
[R9] Bonetto, Riccardo, et al. "Secure communication for smart IoT objects: Protocol stacks, use cases and practical examples." 2012 IEEE international symposium on a world of wireless, mobile and multimedia networks (WoWMoM). IEEE, 2012.
Author Response
In the chapter "Introduction" a description of the evolution of the presented solution was presented (l.156 - l.182) with particular emphasis on a significant new contribution in the article.
In the Chapter 4, subsection 4.3 "Security evaluation" is presented, which describes the resistance of the presented solution to real life attacks.
The content of Chapter 1 has been modified in the area of security measures for the case of IoT in critical areas (l.67-- l.86)
The organization description of the paper was added at the end of the Introduction section (l.183—l.196)
Footnote 1 describes the basic features of the TPM. The manner of using these properties for the needs of the solution is included in the extended description of the concept of protection of sensor nodes in sub-chapter 2.2.
In section 2.2, the description of the concept of solution were extended and in Section 2.3 were added a general description and main tasks of procedures in particular phases of the sensor nodes' domain life. These descriptions should explain more clearly the distribution of symmetric keys in the solution (l.251—l.263).
We were mainly focused on the assessment of the feasibility of the proposed solution and its scalability. For military applications, in addition to security, one of the basic parameters of the applicability of the solution is the time of system preparation and its scalability. We have experimentally determined that the implementation time of secure domain preparation procedures is fully acceptable. We also showed a linear dependence of the number of operations performed, depending on the number of nodes creating the domain. Thus, we showed that the proposed solution is scalable.
The "Related works" section has been expanded (as part of the Introduction). We have included 3 other works related to military applications. The list of publications has been extended by an additional 9 items. We have added items: [3-6], [11-12],[14], [23-24].
Reviewer 3 Report
The authors mentioned that the architecture of WSN is based on a cluster.
If there is no cluster or the dynamic feature is high time variation, could this architecure support a secure demain?
Author Response
The presented solution assumes that sensor nodes are mobile, but operate within the range of the communication link used. Presented solution is not applicable in the case of no cluster or high level variation.
Round 2
Reviewer 1 Report
Authors have improved the paper
Author Response
We would like to express our sincere gratitude for valuable assessments and remarks. It significantly helped us in the manuscript improvement and in choosing the direction of further work as well.
Thank you.
Reviewer 2 Report
I will like to thank the authors for their effort to address the reviewer’s comments. However, there remain some issues that need addressing.
The current form of Introduction section is a combination of Introduction and Related work material. It will more useful to split this section into two, and list all the related works in the second section. There also they should compare the paper at hand with the previous works and justify its novelty.
In the introduction section, a bullet list that defines the contribution of the works is missing.
The evaluation of performance in terms of computational requirements is missing. The authors only provide the communication cost in terms of how many additional bytes/packets the proposed procedures introduce. In the case, that such evaluation is not feasible, they should state that and justify the reasons for such omission.
A minor typo:
l. 941: usedc --> used
Author Response
Response to Reviewer 2 Comments
We would like to express our sincere gratitude for valuable assessments and remarks. It significantly helped us in the manuscript improvement and in choosing the direction of further work as well.
Point 1: The current form of Introduction section is a combination of Introduction and Related work material. It will more useful to split this section into two, and list all the related works in the second section. There also they should compare the paper at hand with the previous works and justify its novelty.
Response 1: The old Introduction section has been divided into two sections Introduction and Related works. Previous work has been described in the final part of Section 2 (marked with a blue background).
Point 2: In the introduction section, a bullet list that defines the contribution of the works is
missing.
Response 2: The contribution of the works is described in the final part of Section 2. (marked with a yellow background).
Point 3: The evaluation of performance in terms of computational requirements is missing. The authors only provide the communication cost in terms of how many additional bytes/packets the proposed procedures introduce. In the case, that such evaluation is not feasible, they should state that and justify the reasons for such omission..
Response 2: An additional subsection 5.4 has been devoted to performance evaluation. (marked with a green background).
Round 3
Reviewer 2 Report
The authors have responded adequately to the reviewer's comments